# VARIABILITY OF NEURAL NETWORKS AND HAN-LAYER: A VARIABILITY-INSPIRED MODEL

## ABSTRACT

What makes an artificial neural network easier to train or to generalize better than its peers? We introduce a notion of *variability* to view such issues under the setting of a fixed number of parameters which is, in general, a dominant cost-factor. Experiments verify that variability correlates positively to the number of activations and negatively to a phenomenon called Collapse to Constants, which is related but not identical to vanishing gradient. Further experiments on stylized problems show that variability is indeed a key performance indicator for fully-connected neural networks. Guided by variability considerations, we propose a new architecture called Householder-absolute neural layers, or Han-layers for short, to build high variability networks with a guaranteed immunity to gradient vanishing or exploding. On small stylized models, Han-layer networks exhibit a far superior generalization ability over fully-connected networks. Extensive empirical results demonstrate that, by judiciously replacing fully-connected layers in large-scale networks such as MLP-Mixers, Han-layers can greatly reduce the number of model parameters while maintaining or improving generalization performance. We will also briefly discuss current limitations of the proposed Han-layer architecture.

## 1 INTRODUCTION

### 1.1 OVERVIEW

Deep neural networks (DNNs) have greatly advanced the state of the arts in many machine learning tasks such as image classification, text categorization, speech recognition, to name just a few out of a long list. Despite their tremendous successes, it remains to be fully understood why DNNs work so well on so many tasks in machine learning. In this paper, we take an intuitive approach to providing a new angle from which one can qualitatively investigate certain behaviors of DNNs and provide explanations to some critical issues. Our new angle is based on a notion of variability for DNNs which seems to have not been specifically examined in the past. Our study of variability focuses primarily on function (forward propagation) values rather than on the gradient (back-propagation) values, offering a different insight from gradient-based views.

In essence, the proposed variability is a qualitative surrogate of expressivity or expressiveness for neural networks (see a recent survey paper (Gühring et al., 2020) and references thereof). Instead of giving a precise, quantitative analysis on what kinds of functions can or cannot be expressed by certain neural network models which in general can be exceedingly difficult, we opt to give a qualitative characterization that, to a degree, reflects the levels of expressivity, thus sidestepping hurdle in analyses while still providing useful information on the expressiveness of neural networks. In doing so, we aim to find useful guidelines for developing new network architectures.

We show that for fully connected and ReLU-activated DNNs with a fixed number of model parameters, the higher variability is (within a reasonable range), the easier it is to train the model to reach high-quality solutions. In particular, variability rises with a quantity called *activation ratio*, or AR (see (5)

for definition), and falls with the occurrence of a phenomenon called Collapse to Constant (C2C), which is not identical (but related) to gradient vanishing. Guided by the variability viewpoint, we propose a novel neural-layer design aimed to increase AR and decrease the chance of C2C at the same time.

## 1.2 MAIN CONTRIBUTIONS

The purpose of this work is to gain more insights into the behaviors of DNNs and then use them to build new DNN models. We study the notion of variability that reflects DNN's expressivity and trainability. Guided by variability, we construct a new architecture called Householder-absolute neural layer or network (Han-layer or HanNet) to attain higher variability with fewer parameters. As the name suggests, in a fully connected layer or network (FC-layer or FCNet), we replace square weight matrices with Householder reflectors and use the absolute-value function for activations. To sum up, our contributions are both conceptual and practical, consisting of the following aspects.

We study the notion of variability, along with a measurement for it, for DNNs that adds a new angle to view and explain the behaviors of DNNs. We construct HanNet to achieve a high variability and at the same time guarantee an immunity to vanishing or exploding gradient. The chance of Collapse to Constants, yet not completely eliminated in theory, has also been greatly diminished.

On stylized small problems, our experiments suggest that variability is indeed a key performance indicator for ReLU-activated FCNets and that HanNets possess an unusually high level of generalization ability. As is illustrated in Figure 4 in Section 5.2.1, a HanNet outperforms FCNets by a huge margin, producing nearly perfect results.

On several standard datasets in regression and image classification, our experiments indicate that comparing to FCNets, HanNets can greatly reduce the number of model parameters, sometimes by orders of magnitude, while still maintaining and often improving generalization performance. This property represents a promising feature with potential impacts on large-scale applications.

Many important issues remain to be investigated in order to better understand the power and the limits of HanNets, and to unlock their potentials in real-world applications.

## 2 RELATED WORK

As is well known, initialization is a critical step towards the success or failure of neural network training. We start our investigation by visualizing landscapes of DNNs with properly initialized parameters with different activation functions. Landscapes of DNNs have been examined from various angles and for different purposes, for example see (Kawaguchi, 2016; Li et al., 2017; Laurent & Brecht, 2018; Kuditipudi et al., 2019; Fort et al., 2020; Sun et al., 2020). In our case, visualization results lead to the concept of variability. The two key building blocks in our proposed HanNets are (1) the absolute-value (ABS) function as an activation function and (2) the Householder reflectors as the orthogonal weight matrices — two topics that previously appeared in literature within quite different contexts from ours; for example, ABS functions in (Batruni, 1991; Lin & Unbehauen, 1992; Karnewar, 2018) and Householder matrices in (Mhammedi et al., 2017; Tomczak & Welling, 2016; Vorontsov et al., 2017; Wang et al., 2020; Wisdom et al., 2016; Zhang et al., 2018). In addition, other related works will be referenced throughout the paper.

### 2.1 NOTATIONS

Given an activation function $\phi(\cdot)$ and a sequence of weight-bias parameters $\{(W_i, b_i)\}$ of suitable sizes, we first define layer functions

$$\psi_i(\cdot) \equiv \psi_i(\cdot, W_i, b_i) := W_i\phi(\cdot) + b_i, \ i = 1, 2, \cdots, \tag{1}$$

each of which is the composition of an affine function with the activation $\phi(\cdot)$. As mentioned above, we often drop the dependence of $\psi_i$ on the parameter pair $(W_i, b_i)$ whenever no confusion arises. For

any positive integer $k$, define

$$F_k(x) := (\psi_k \circ \cdots \circ \psi_1)(x), \tag{2}$$

which is the composition of $\psi_1, \ldots, \psi_k$, parameterized by the weight-bias pairs $(W_i, b_i)$ for $i = 1, \cdots, k$. In general, $W_i \in \mathbb{R}^{n_{i+1} \times n_i}$, $b_i \in \mathbb{R}^{n_{i+1}}$ and $\phi_i : \mathbb{R}^{n_{i+1}} \to \mathbb{R}^{n_{i+1}}$, thus function $F_k(x)$ is from $\mathbb{R}^{n_1}$ to $\mathbb{R}^{n_k}$. For an $L$-layer DNN, the model output is $F_L(x, \mathbf{W}, \mathbf{b})$, where $\mathbf{W} = \{W_1, ..., W_L\}$ and $\mathbf{b} = \{b_1, ..., b_L\}$. For any given parameter pair, the network maps an input $x$ to an output $F_L(x, \mathbf{W}, \mathbf{b})$, which can be computed through the forward propagation:

$$s_0 = x; \;\; z_k = W_k s_{k-1} + b_k, s_k = \phi(z_k), \;\; k = 1, \cdots, L. \tag{3}$$

## 3 RISE AND FALL OF VARIABILITY

We start with a set of simple experiments in Section 3.1 to observe the landscapes of neural network functions $F_L(x)$ as the network depth $L$ increases while the total number of model parameters is kept approximately unchanged.

### 3.1 LANDSCAPE EXPERIMENTS

To facilitate visualization, we add an input layer and an output layer, both of dimension 2, to the $L$ hidden layers. For convenience, we continue to use $F_L(x)$ to denote the extended network. The data set is a set of grid points on the square $[-1, 1]^2 \subset \mathbb{R}^2$ over which we compute and plot the surface $z = \|F_L(x, \mathbf{W}, \mathbf{b})\|^2$ for randomly sampled $(\mathbf{W}, \mathbf{b})$ with proper scalings. Three activation functions, Sigmoid $\phi(t) = 1/(1 + e^{-t})$, ReLU $\phi(t) = \max(0, t)$ and ABS $\phi(t) = |t|$, are used. For ReLU we use Kaiming initialization, and for other two functions we use Xavier initialization (Glorot & Bengio, 2010; He et al., 2015).

We keep the total number of parameters in $(\mathbf{W}, \mathbf{b})$ at around 10,000. For each given depth value $L$, we calculate the corresponding width $d$ (rounded to the nearest integetr), then compute and plot $z$ over the grid on $[-1, 1]$ for 5 different samples for $(\mathbf{W}, \mathbf{b})$. Figure 1 is the result of one single random sample, while the complete plots are given in Appendix (see Figures 12, 13 and 14, where each row consists of 5 plots corresponding to 5 random parameter samples).

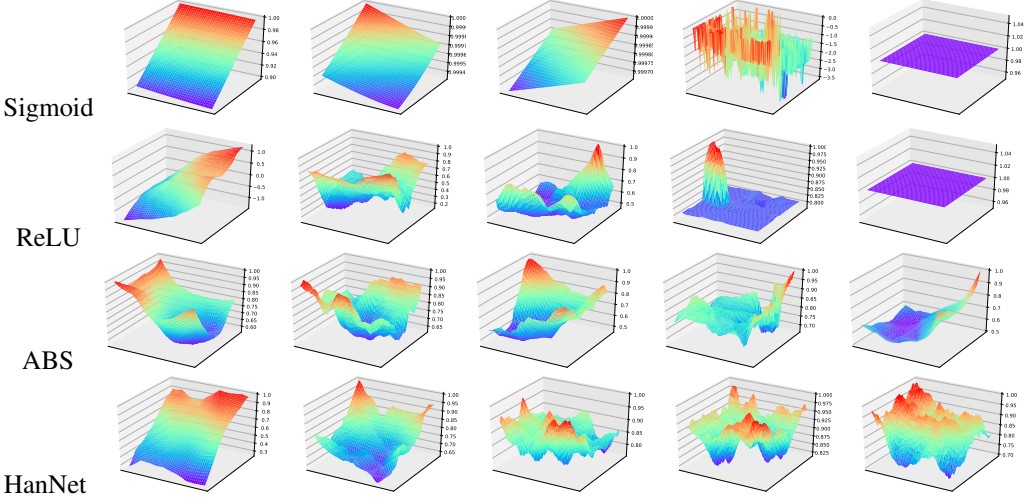

Figure 1: Landscape of $\|F_L(x, \mathbf{W}, \mathbf{b})\|^2$. Rows 1 to 3 are for FCNets with 3 activations: Sigmoid ($L = 2, 4, 6, 10, 15$), ReLU and ABS ($L = 2, 10, 20, 40, 60$). Row 4 is for HanNets (to be defined) with the same $L$ values as for ReLU and ABS.

Based on these plots, we make the following observations. For Sigmoid, the surfaces are rather monotonous with few variations either in the $x$-space (within each plot) or in the parameter space

(across the 5 plots). In particular, in the $x$-space, we hardly see any peaks and valleys. Most remarkably, with slightly larger $L$ values, the surfaces become constants. We call this curious phenomenon "Collapse to Constants" or simply C2C. For ReLU, the landscapes have much richer expressions from the very beginning ($L = 2$). The amount of variations increases as $L$ grows deeper from 2 to 10 and even to 20. However, at $L = 40$ and 60, the phenomenon of C2C again shows up. ABS Activation seems more C2C-resistive than ReLU. Many peaks and valleys still exist at deep depths ($L = 40, 60$).

Upon further examinations, it is clear that for large $L$ not only the scalar function $\|F_L(x, \mathbf{W}, \mathbf{b})\|$ tends to constants in $x$-space, but in fact the vector-valued function $F_L(x, \mathbf{W}, \mathbf{b})$ itself tends to constant vectors for all $x \in [-1, 1]^2$. We will explain this C2C phenomenon later.

## 3.2 Variability and a Measurement for it

From the above experimental results, we see unmistakable differences in the outputs of network functions $F_L(x, \mathbf{W}, \mathbf{b})$ from one activation function to another and as the depth $L$ grows. All the differences are rooted in the richness of landscape variations in the data space where $x$ varies, as well as in the parameter space where $(\mathbf{W}, \mathbf{b})$ varies. We will refer to such richness of landscape variations as *variability* of network functions.

Intuitively speaking, higher variability means not only higher expressiveness in data space, but also better responsiveness to parameter changes. On the other hand, a network with low variability means that it may have a poor approximation power and consequently may be difficult to train. In particular, a deep network would have lost all variability, in certain regions of the parameter space, when it becomes a constant in the entire data space for parameters from those regions, which we refer to as Collapse to Constant (C2C). In Appendix A.4, we treat C2C and its characteristics with more details. Variability may serve as a predictive indicator on trainability of suitable DNNs. We present numerical evidence in Appendix B, verifying that the pattern of variability is highly correlated to the performance of corresponding DNNs.

In general, it is nontrivial to develop effective and computable measures for variability, which should be a subject of investigations on its own right. Here we present a variability measurement that seems to have worked well with two-dimensional data spaces. We define

$$\mathrm{V}_3 := \mathbb{E}_{(\mathbf{W}, \mathbf{b})} \left( \|f\|_\infty^{-1} \int_\Omega \left\| \frac{\Delta^3 f}{\Delta x_i^3} \right\|_1 dx \right) \tag{4}$$

where $f \equiv f(F_L(x; \mathbf{W}, \mathbf{b}))$ is a scalar-valued function, the mean is taken over a certain region of interest in the parameter space, and $\Delta^3 f / \Delta x_i^3$ is a finite-difference approximation of the third derivative over a grid. The phenomenon of C2C occurs when $V_3$ is close to zero. For example, if $f$ is a least squares loss function and $F_L(\cdot)$ is linear, then $V_3$ vanishes. On the other hand, when $F_L(\cdot)$ has high nonlinearity, then the value $V_3$ will be relatively large.

In Figure 2, we plot the variability of $F_L(x; \mathbf{W}, \mathbf{b})$ on $\Omega = [-1, 1]^2$, computed as described in Appendix A.2, for network depth $L$ varying from 3 to 45 (with increment 3). As we see on the left plot for ReLU, the $V_3$-variability initially increases with $L$ until it reaches its peak at around $L = 16$. Afterwards, it starts to decline until it vanishes suddenly at $L = 25$ because the network output becomes constant at least at one parameter sample, making the geometric mean zero. On the middle plot for absolute-value, besides a slightly higher profile, after the peak variability declines but does not vanish up to $L = 60$.

## 3.3 Rise with Activation Ratio and Fall with C2C

It should be clear that variability comes from nonlinear activations in the model. As we see from the previous variability experiments, variability initially always rises as the network depth increases until it reaches a peak. There is a simple explanation for this. That is, when the total number of parameters is fixed, the number of activations always increases with the depth. The *activation ratio* (AR) for a

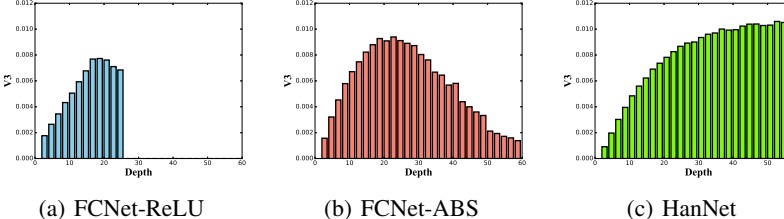

| (a) FCNet-ReLU | (b) FCNet-ABS | (c) HanNet |

Figure 2: Variability measured by $V_3$ in (4) for FCNet-ReLU (left) and FCNet-ABS (middle) and HanNets (right). Each bar represents a geometric mean of 3000 parameter samples. As depth $L$ grows, the width $d$ decreases so that the total number of model parameters is approximately 4000. The structure of HanNets is the same as that of FCNets with far fewer model parameters.

model is defined below

$$\text{AR} = \frac{total\ number\ of\ individual\ activations}{total\ number\ of\ model\ parameters}. \tag{5}$$

The activation ratio is monotonically increasing with depth $L$. In particular, the rate of increase is the largest at $L = 1$, which shows that the activation ratio increases most rapidly at the beginning, explaining why variability rises quickly at the very beginning, see Appendix A.3 for a formula and more details. In this work, we will also use AR as one of the metrics for variability. However, we caution against the simplistic view that the higher the AR or variability is, the better. In fact, experiments indicate that excessive nonlinearity tends to increase training difficulties.

It should be clear that variability declines and eventually vanishes as C2C develops and materializes. A detailed treatment, including a quantitative characterization of C2C, is given in Appendix A.4.

## 4 HANNET: A HIGH VARIABILITY MODEL

A path to enhancing and maintaining variability of a DNN is to raise its resistance level to C2C and increase its AR simultaneously. Our construction of Han-layers follows exactly this path.

### 4.1 HOUSEHOLDER WEIGHTING: ORTHOGONAL AND HIGH AR

It is well-known that Householder reflection matrix associated with a nonzero vector $u \in \mathcal{R}^n$ is

$$H(u) = I - 2uu^{\mathsf{T}}/\|u\|^2, \tag{6}$$

which is symmetric and orthogonal. As is well known, orthogonality is a desirable property for weight matrices in DNNs. In addition, the degree of freedom in $H(u)$ is $n$, one order of magnitude smaller than $O(n^2)$ for a generic orthogonal matrix.

Evidently, the activation ratio of a DNN can be increased by reducing the number of model parameters without changing the network depth and width, such as by parametrizing an $n \times n$ weight matrix with far less than $O(n^2)$ parameters (similar to what is done in CNNs). This motivates us to use a Householder reflection matrix to replace a general matrix or a generic orthogonal matrix in a neural layer so that the AR value associated with this layer is enlarged by a factor of $O(n)$. In natural language processing areas, it has been proposed to use a product of multiple Householder matrices to parameterize weight matrices in recursive neural networks, for example (Mhammedi et al., 2017).

### 4.2 HAN-LAYERS AND HIGH VARIABILITY

A Householder-absolute neural layer, or Han-layer, is composed of a Householder matrix followed by ABS activation (see Appendix C.1 for an algorithmic form), and a HanNet structure is created by using multiple Han-layers. We have already visualized the landscape of a deep HanNet in Figure 1, which has more peaks and valleys than the others implying a higher level of variability. In Figure

2, we calculate the variability measure $V_3$ defined in (4), where the geometric mean is used that vanishes whenever C2C occurs. We observe that the variability of HanNets remains at a high level without deterioration in the entire tested range.

### 4.3 IMMUNITY TO GRADIENT VANISHING OR EXPLODING

Let the neural network function $F_L(x, \mathbf{W}, \mathbf{b})$ be defined in (2). The spectrum of a so-called $G_L$-matrix (see (12) in Appendix A.4) characterizes whether vanishing or exploding gradient would happen or not. Roughly speaking, $G_L$-matrix is the dominant part of the gradient. As $L \to \infty$, vanishing gradient corresponds to $G_L \to 0$ and exploding gradient to $G_L \to \infty$. For HanNet, with the notation $\mathbf{u} = \{u_k\}_{k=1}^L$, the $G_L$-matrix takes the form

$$G_L(x, \mathbf{u}, \mathbf{b}) = \prod_{k=1}^L H(u_k) \nabla \phi(z_k), \tag{7}$$

where $H(u_k)$ is defined by (6) and $\nabla \phi(z_k)$ is the diagonal, Jacobian matrix of $\phi(t) = |t|$ evaluated component-wise at a vector $z_k$. Evidently, the diagonal matrices $\nabla \phi(z_k)$ are, with a high probability, all orthogonal matrices with diagonal entries equal to $\pm 1$. Hence, the following result holds implying that gradient can never vanish or explode (in a probability sense).

**Proposition 1.** *Under a mild distribution assumption, the G-matrix for HanNet, that is, $G_L(x, \boldsymbol{u}, \boldsymbol{b})$ defined in 7, remains orthogonal with probability one for any $x$, any $u_i \neq 0$ in $\boldsymbol{u}$, any $\boldsymbol{b}$, and any integer $L > 0$.*

### 4.4 HAN/MLP-MIXER

In literature, the term Multi-Layer Perceptron (MLP) is often used exchangeably with FCNet. Recently, MLP-dominated models have seen a wave of revivals for image recognition tasks (Tolstikhin et al., 2021; Liu et al., 2021). MLP-dominated models (without multi-head attentions) are much more concise than Transformer-based models (Dosovitskiy et al., 2020; Touvron et al., 2021) but can still maintain testing performances on very large-scale datasets. The motivation of MLP-Mixer (Tolstikhin et al., 2021) is to use the purely fully-connected layers to remove attention architectures. An MLP-Mixer block is the elementary unit in MLP-Mixer models that consists of several FC-layers and skip-connections to form the following map from input $X$ to output $Y$,

$$Z = X + \textbf{GELU}\left(W_2 \, \textbf{Layer Norm}(W_1 X)\right), \tag{8}$$

$$Y = Z + \textbf{GELU}\left(\textbf{Layer Norm}(Z W_3) \, W_4\right), \tag{9}$$

where the first row (8) is called token-mixing for cross-token communication and the second row (9) is called channel-mixing for cross-channel communication, both being of MLP structure. Here we form our Han-Mixer block by replacing all weight matrices $W_i$ by Householder matrices $H_i$, $i = 1, 2, 3, 4$, and all activation functions by the absolute function **ABS**, that is,

$$Z = \textbf{ABS}(H_2 \textbf{ABS}(H_1 X)), \quad Y = \textbf{ABS}(\textbf{ABS}(Z H_3) H_4), \tag{10}$$

where we remove skip-connections and layer-normalizations (since HanNets does not suffer from gradient problems). See Appendix C.2 for figurative illustrations of MLP-Mixer vs. Han-Mixer blocks. The resulting Han/MLP-Mixer models are shown in Figure 3, where we arrange some Han-Mixer blocks after MLP-Mixer blocks (which may be empty). The main reason for us to combine the two types of Mixer blocks is that, short of drastically increasing network width, Han-Mixers alone cannot always provide enough model parameters for large-scale datasets. In addition, we use the convolution stem recommended by (Xiao et al., 2021) instead of the one in (Tolstikhin et al., 2021).

## 5 EXPERIMENTS

### 5.1 DATASETS

To empirically investigate the efficacy of HanNets in comparison to FCNets, we use multiple datasets including a synthetic dataset Checkerboard, 2 classic regression datasets, Elevators (Dua

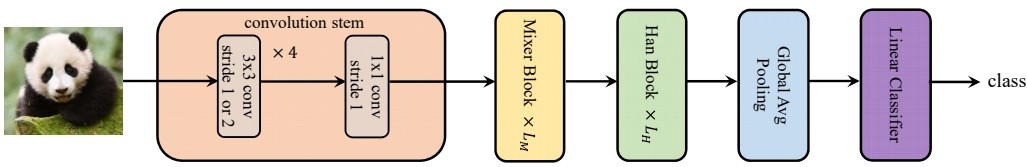

Figure 3: Overall structure of the tested Han/MLP-Mixer model ($L_M$ can be zero).

& Graff, 2017) and Cal Housing (Pace & Barry, 1997), and 4 widely used image classification datasets CIFAR10, CIFAR100, STL10, and ImageNet32 (Krizhevsky et al., 2009; Coates et al., 2011; Chrabaszcz et al., 2017), where the last one is a down-sampled version of ImageNet (out of affordability considerations). All dataset settings and training details are given in Appendix D.1.

## 5.2  RESULTS

### 5.2.1  SYNTHETIC DATASET: CHECKERBOARD

The ground truth and the training set, along with a couple of typical results, are plotted in Figure 4.

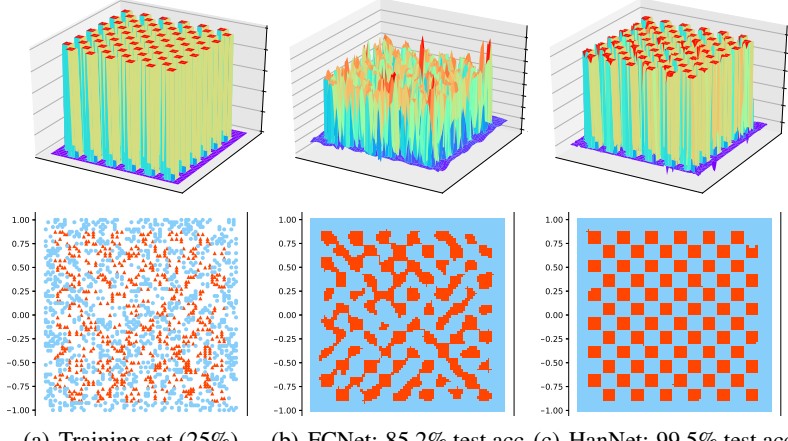

(a)  Training set (25%)    (b)  FCNet: 85.2% test acc  (c)  HanNet: 99.5% test acc

Figure 4: The top row consists 3 functions (from left to right): the target binary function, a function trained from a FCNet and one from a HanNet, respectively. The bottom row: (a) the training set with 25% of data points, (b) and (c) are top views of the two trained solutions (rounded to 0 or 1), along with their test accuracies. The training accuracies are nearly 100% for both.

On this dataset, we compare a range of FCNets with HanNets (in fact, ResNets were also compared, without better results than those from FCNets). We generate more than 200 pairs of FCNets and HanNets with width varying from 20 to 100 (increment 10) and depth from 2 to 30. Figure 5 (a, b) gives the test results in the heat-map form. We observe from Figure 5 that overall there exists a large gap in testing accuracy between the FCNets (around 85%) and HanNet (over 99%) over a wide range of test cases. The unexpected, near-perfect results obtained by HanNets, without any explicit regularization, look quite stunning and are stable in the sense that similar results can be trained from other random 25% vs. 75% data-splittings. We observe that over-parametrization is not a necessary condition for near perfect solutions. In fact, 400 parameters are sufficient to achieve near-zero testing error in one of the cases in Figure 5(b).

An ablation study in Appendix D.2 suggests that Householder weighting and ABS activating be equally critical to the surprising performance of HanNets on the Checkerboard dataset. In addition, A stability study in Appendix D.3 is done in which a portion of training labels is randomly flipped, and the results show that with seriously damaged labels HanNets still maintain a clear advantage over

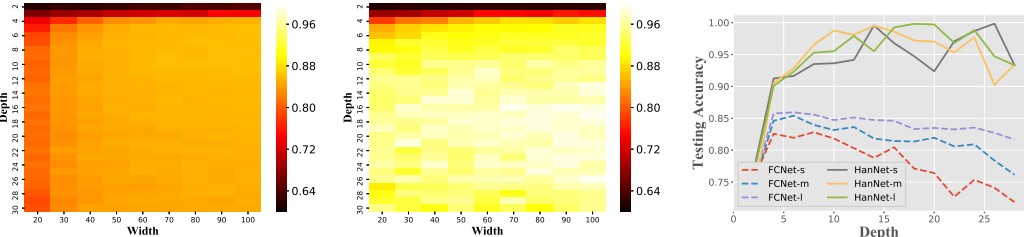

Figure 5: (Left) FCNet: the best testing accuracy is up to 87%. (Mid) HanNet: the best testing accuracy is over 99%. (Right) FCNet vs HanNet: the number of model parameters in each line is fixed, respectively, at 4000, 8000 and 20000 for FCNet-s, FCNet-m and FCNet-l, and at 1000, 3000, 5000 for the 3 lines for HanNets. In all experiments, the training errors are near zero. The plot shows the existence of significant gaps in generalization ability between FCNets and HanNets.

FCNets in testing performance. In addition, extensive experiments show that variability is closely and positively related to trainability (see Appendix B for details).

### 5.2.2 REGRESSION DATASETS: ELEVATORS AND CAL HOUSING

On the two regression Datasets, we compare the performance of a HanNet with two FCNets. Table 1 lists the relevant statistics for the 3 DNNs where depth refers to the number of hidden layers (there exist additional, data-size-dependent input/output layers). We see that in terms of parameter numbers FCNet1 and HanNet are comparable peers, while FCNet2 has about 15 times more parameters.

| Model | FCNet1 | FCNet2 | HanNet |
|---|---|---|---|
| Depth × Width (#param) | $5 \times 50$ (10.9K) | $5 \times 200$ (165K) | $20 \times 200$ (10.6K) |

Table 1: Essential statistics of DNNs compared (more details in Appendix D.4).

We present testing NRMSE (normalized root mean square error) loss values in Figure 11 in Appendix D.4, where the curves are average values over 5 trials with error bars. Clearly, HanNet outperforms its "peer" FCNet1 by a notable margin, especially when fewer training samples are used, and is competitive with (on 80% training data) or better than (on 20% training data) FCNet2 which uses 15 times more parameters. We mention that the best testing performance of HanNet is statistically the same as that reported in (Tsang et al., 2018) with an FCNet model that uses over 5 million parameters. Additionally, with fewer parameters, HanNet appears less influenced by overfitting.

### 5.2.3 IMAGE CLASSIFICATION DATASETS

In this section, we investigate HanNets' generalization ability in image classification by comparing MLP-Mixers with our Han/MLP-Mixers presented in Subsection 4.4. The comparisons are done with 3 standard image datasets and a down-sampled version of ImageNet (due to computing source limitation). More experimental details are given in Appendix D.5.

Empirical evidences from MLP-based models (Tolstikhin et al., 2021) suggest that these models can achieve the state-of-the-art results on very large-scale datasets. Since the datasets used in our tests are not large enough to play to the full strengths of Mixer models, our aim here is not to observe how close our results can approach the state-of-the-art, but how Han-layer structures can impact the performance of Mixer models.

Tables 2 and 3 report test results from various MLP-Mixer and H/M-Mixer models. On the smaller dataset STL10, HanMixers alone can already produce the best result. On larger datasets CIFAR10 and CIFAR100, enhanced performance from HanMixers are achieved either by adding a relatively few parameters (e.g., MLPMixer (2 or 4) vs MLPMixer (2 or 4) + HanMixer (16)), or even with

|  | #Param | STL10 | CIFAR10 | CIFAR100 |
|---|---|---|---|---|
| CNN stem | 1.82 M | 18.2 % | 7.2 % | 27.8 % |
| +MLPMixer (2) | 2.89 M | 18.2 % | 5.3 % | 27.2 % |
| +MLPMixer (4) | 3.96 M | 18.7 % | 5.5 % | 27.1 % |
| +MLPMixer (0) + HanMixer (16) | 1.86 M | **15.3** % | 5.9 % | 26.7 % |
| +MLPMixer (1) + HanMixer (16) | 2.39 M | 16.7 % | 5.0 % | 24.6 % |
| +MLPMixer (2) + HanMixer (16) | 2.93 M | 17.1 % | 4.7 % | **24.4** % |
| +MLPMixer (4) + HanMixer (16) | 4.00 M | 17.8 % | **4.6** % | 24.7 % |

Table 2: Error rates (%) on CIFAR and STL10 datasets.

|  | #Param | Top1 error | Top5 error |
|---|---|---|---|
| CNN stem | 8.28 M | 55.7 % | 31.2 % |
| +MLPMixer (4) | 16.7 M | 44.2 % | 21.2 % |
| +MLPMixer (8) | 27.2 M | 42.6 % | 20.2 % |
| +MLPMixer (0) + HanMixer (16) | 8.35 M | 48.9 % | 26.1 % |
| +MLPMixer (4) + HanMixer (16) | 16.8 M | **41.1** % | **19.2** % |
| WideResNet (Chrabaszcz et al., 2017) | 37.1 M | **41.0** % | **18.9** % |

Table 3: : The top-1 and top-5 error rates on ImageNet32. On this dataset, among MLPMixer ($i$) for $i = 4, 8, 12, 16$, $i = 8$ gives the best results,

far fewer parameters (e.g., MLPMixer (4) vs MLPMixer (1)+HanMixer (16)). On ImageNet32, our Han/MLP-Mixer combination model clearly outperforms pure MLPMixers and offers a competitive performance with WideResNet while using only 40% parameters. In summary, the benefits of using HanMixers have been made evident in this set of experiments.

## 6 CONCLUSIONS

As a qualitative surrogate for expressivity and trainability, variability provides a useful angle to view certain critical behaviors of DNNs. Empirical evidence suggests that activation ratio and Collapse to Constants are two major contributing factors to the gain and loss of variability, respectively. Based on such insights, we propose a Han-layer architecture consisting of Householder weighting and absolute-value activating. This new architecture greatly raises activation ratio and provides a guaranteed immunity to vanishing or exploding gradient.

Extensive experimental results indicate that when used judiciously as replacements of fully connected layers, Han-layers can significantly reduce the number of model parameters (thus potentially the associated computing costs), while maintaining or improving the level of generalization performance, as is shown by our results on MLP-Mixer models for image classification.

On the synthetic Checkerboard dataset, HanNets consistently exhibit a surprisingly high level of generalization ability. It remains to be investigated that under what conditions and to what extent such an ability could be realized on large-scale datasets.

Several limitations in our results are worth mentioning. So far, our measurement of variability is limited to low-dimensional data spaces. It remains to be seen whether or not a computable extension to high-dimensional spaces can be fruitfully done. As is currently defined, a Han-layer is not as flexible as a fully connected layer since a Householder matrix must be square. Finally, we offer two points of perspectives: (1) measured variability should be viewed within a reasonable range (chaotic landscapes are generally bad); and (2) too many Han-layers, even though immune to gradient problems, may still increase the difficulty in training due to high nonlinearity.

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

# A VARIABILITY

## A.1 EXISTING TECHNIQUES THAT ENHANCE VARIABILITY

Several existing techniques in deep learning can be interpreted from the viewpoint of enhancing the variability of neural networks. For example, convolutional neural networks (CNN) use far fewer parameters, in comparison to fully connected networks, at each layer, thus significantly increasing activation ratios and subsequently enhancing variability. In this view, achieving higher activation ratios should be a significant contributing factor to the great success of CNN.

Since C2C has a close relationship with vanishing gradient, it is not surprising that existing techniques designed to alleviate the latter can also help with the former. Specifically, since G-matrices are limits of C-matrices, techniques that slow down the size decrease of G-matrices usually also slow down the size decrease of C-matrices. Such techniques include Residual Networks (or ResNet) (He et al., 2016) and Batch Normalizations (Ioffe & Szegedy, 2015). A particularly simple and widely utilized technique is to initialize weight matrices by orthogonal matrices (Hu et al., 2019; Huang et al., 2018).

## A.2 SETTINGS OF $V_3$ IN FIGURE 4

In our experiment, we use the least-squares function: $f(\cdot) := \| \cdot \|_2^2$ and $\Omega = [-1, 1]^2$. The mean over parameters will be computed as a sample mean. In particular, we use the geometric mean which vanished if a single sampled value is zero. Therefore, once $F_L(x; \mathbf{W}, \mathbf{b})$ becomes a constant (or linear for that matter) on $\Omega$, variability as measured by $V_3$ vanishes.

## A.3 INCREASE OF ACTIVATION RATIO

For simplicity, let us consider fully connected (FC) network of $L$ layers for which the number of parameters in $(\mathbf{W}, \mathbf{b})$ is $N_w = (d^2 + d)L$ where $d$ is the width of the layers. In this case, the number of activations is $Ld$. The *activation ratio* (AR) $\rho$ for this network is the total number of activations divided by the total number of parameters, i.e., $\rho = Ld/N_w$, which can be written as a function of $L$. By solving the equation $N_w = (d^2 + d)L$ for $d$, we obtain $d = (1 + \sqrt{1 + 4N_w/L})/2$. Therefore, the activation ratio is

$$\rho(L) = \frac{1}{N_w} \left( L + \sqrt{L^2 + 4LN_w} \right), \quad L \geq 1. \tag{11}$$

Furthermore, since

$$\rho'(L) = \frac{1}{N_w} \left( 1 + \frac{1 + 2N_w/L}{\sqrt{1 + 4N_w/L}} \right) > 0,$$

the activation ratio is monotonically increasing with $L$. In particular, the rate of increase is the largest at $L = 1$. This simple calculation shows that the activation ratio increases most rapidly at the beginning, explaining why variability arises quickly at the very beginning.

## A.4 COLLAPSE TO CONSTANTS AND ITS CHARACTERIZATION

To compute the derivative of $F_L(x, \mathbf{W}, \mathbf{b})$ with respect to the parameters, one uses the chain-rule to obtain so-called back-propagation formulas, such as

$$\frac{\partial}{\partial b_1} F_L(x, \mathbf{W}, \mathbf{b}) = \frac{\partial s_L}{\partial b_1} = \nabla\phi(z_1)W_2^T \nabla\phi(z_2) \cdots W_L^T \nabla\phi(z_L),$$

where $z_k$ are computed in (3), $\nabla\phi(z_k)$ are diagonal matrices with scalar-valued $\phi'$ applied component-wise. It is well-known that the behavior of the derivatives is critically determined by the properties of the above matrix product. For convenience and without loss of generality, we add $W_1$ to the product and define

$$G_L \equiv G_L(x, \mathbf{W}, \mathbf{b}) := \prod_{k=1}^{L} W_k^T \nabla\phi(z_k), \tag{12}$$

which we will simply call the $G$-matrix associated with the network $F_L(x, \mathbf{W}, \mathbf{b})$. It is well known in deep learning that excessively small (or large) size of $G_L$ causes vanishing (or exploding) gradient, which is a major source of difficulty in training.

Now we define another matrix product called the $C$-matrix, by replacing the derivative $\nabla \phi(z_k)$ in (12) by the finite difference at two points $z_k$ and $\bar{z}_k$; that is,

$$C_L \equiv C_L(x, \bar{x}, \mathbf{W}, \mathbf{b}) := \prod_{k=1}^{L} W_k^T D_\phi(z_k, \bar{z}_k), \tag{13}$$

where $D_\phi(\cdot, \cdot) \in \mathbb{R}^{d \times d}$ is diagonal defined by

$$[D_\phi(u, v)]_{ii} = \frac{[\phi(u) - \phi(v)]_i}{[u - v]_i}, \quad i = 1, \cdots, d, \tag{14}$$

with the convention $0/0 = 1$, $\{z_k\}$ and $\{\bar{z}_k\}$ are computed via the recursion (3) starting from $x$ and $\bar{x}$, respectively. By their definitions, it is clear that G-matrices are limits of C-matrices.

The following proposition shows that C-matrices characterize the C2C phenomenon.

**Proposition 2.** *Let network $F_L(x, \mathbf{W}, \boldsymbol{b}) : \mathbb{R}^d \to \mathbb{R}^d$ be defined as in (2). For any two distinct points $x, \bar{x} \in \mathbb{R}^d$, there holds*

$$F_L(x) - F_L(\bar{x}) = [C_L(x, \bar{x})]^T (x - \bar{x}). \tag{15}$$

*Consequently,*

$$\lim_{L \to \infty} C_L(x, \bar{x}) = 0 \implies \lim_{L \to \infty} (F_L(x) - F_L(\bar{x})) = 0. \tag{16}$$

The verification of this proposition is straightforward, so we omit it.

We note that the difference going to zero in (16) does not imply that each individual sequence goes to the same limit. On the contrary, such limits generally do not exist if the bias sequence $\{b_k\}$ is bounded away from zero.

Regarding the C2C phenomenon, the following remarks are in order.

- Wherever $C_L(x, \bar{x})$ is sufficiently small, the output values of the network for these two inputs $x$ and $\bar{x}$ will be close to each other for large $L$.
- $C_L(x, \bar{x})$ will be small if, for all or many $k$, $\|W_k^T D_\phi(z_k, \bar{z}_k)\|$ are sufficiently smaller than 1 in some norm which occurs when one or both of the matrices is sufficiently small.
- If $C_L(x, \bar{x})$ is sufficiently small for all $x$ in some region, then the corresponding outputs of the network will be like a constant in this region. In particular, this can happen when weight matrices in $\{W_k\}$ are small for all or many $k$.

If the weight matrices $W_k, k = 1, \cdots, L$, are properly normalized (for example, all $W_k$ are orthogonal matrices), then the size of $C_L$ will be solely determined by that of $D_\phi(z_k, \bar{z}_k)$ for a given point pair $(x, \bar{x})$, which in turn depends on activation $\phi$ in use. We consider the popular ReLU activation $\phi(t) = \max(0, t)$ and, again for the purpose of contrast, the absolute-value function $\phi(t) = |t|$. In both cases, the diagonal entries in (14) satisfy $[D_\phi(u, v)]_{ii} \in [-1, 1]$.

**Proposition 3.** *Suppose that $u, v \in \mathbb{R}^d$ be i.i.d. random variables with*

$$\mathbf{Prob}([u]_i \geq 0) = \mathbf{Prob}([v]_i \geq 0) = p \in (0, 1), \quad i = 1, 2, \cdots, d.$$

*Then*

$$\mathbf{Prob}\left(|\phi(u) - \phi(v)| = |u - v|\right) = \begin{cases} p^{2d}, & \phi(t) = \max(0, t) \\ \left(p^2 + (1 - p)^2\right)^d, & \phi(t) = |t| \end{cases} \tag{17}$$

*where the absolute values are taken component-wise.*

*Proof.* Consider the scalar case $d = 1$ with $u \neq v$. For ReLU function $\phi(t) = \max(0, t)$,

$$\left| \frac{\phi(u) - \phi(v)}{u - v} \right| \begin{cases} = 1, & u, v \geq 0 \\ < 1, & \text{otherwise} \end{cases}$$

where the probability for the first case (ratio equal to 1) is $p^2$. For absolute value $\phi(t) = |t|$,

$$\left| \frac{\phi(u) - \phi(v)}{u - v} \right| \begin{cases} = 1, & uv \geq 0 \\ < 1, & \text{otherwise} \end{cases}$$

where the probability for the first case (ratio equal to 1) is $p^2 + (1 - p)^2$.

Since all the components are i.i.d., by raising the above probabilities to their $d$-th power, we obtain the corresponding probabilities for the vector case for $d > 1$. $\square$

The proposition indicates that the probability for ReLU to preserve distances in $\mathbb{R}^d$ is much smaller than that for absolute-value. For instance,

- when $p = 1/2$ the above two probabilities in (17) become $1/4^d$ and $1/2^d$, respectively; this is, the latter is $2^d$ times larger than the former;

- when $p = 1/4$ the above two probabilities in (17) become $(1/16)^d$ and $(10/16)^d$, respectively; this is, the latter is $10^d$ times larger than the former.

**Remark 4.** *When $d$ is large, the probability is small for either function to maintain diagonals $|[D_\phi(u, v)]_{ii}| = 1$ for all indices $i$. That is, probabilistically speaking, the two functions are contractive, i.e., with a high probability, they shrink the distance between points after each application, while ReLU is a more forceful contraction than absolute-value is, especially when the positive probability $p$ of input is small.*

In Figure 6, we present a couple of typical experimental results on C- and G-matrices. The left plot is for ReLU activations only with random weight matrices and bias vectors specified, then multiplied by the Xavier/Kaiming initialization constant $\sqrt{2/d}$. For a randomly select point pair $(x, \bar{x})$, we compute the corresponding C-matrix, as well as the G-matrices at these two points, for depth $L$ from 1 to 1000. The curves depict the spectral norms of the matrices $C_L$ and $G_L$ (two of those) at each depth value $L$. As we can see, although for $L$ three curves roughly follow a similar pattern of up-and-downs, the C-matrix still differs significantly from the two G-matrices in that it becomes about two-orders of magnitude smaller than the two G-matrices at the same depth. This signifies an important point that, at least in some cases, C2C should be considered as the leading cause for loss of trainability, instead of vanishing gradient as commonly perceived.

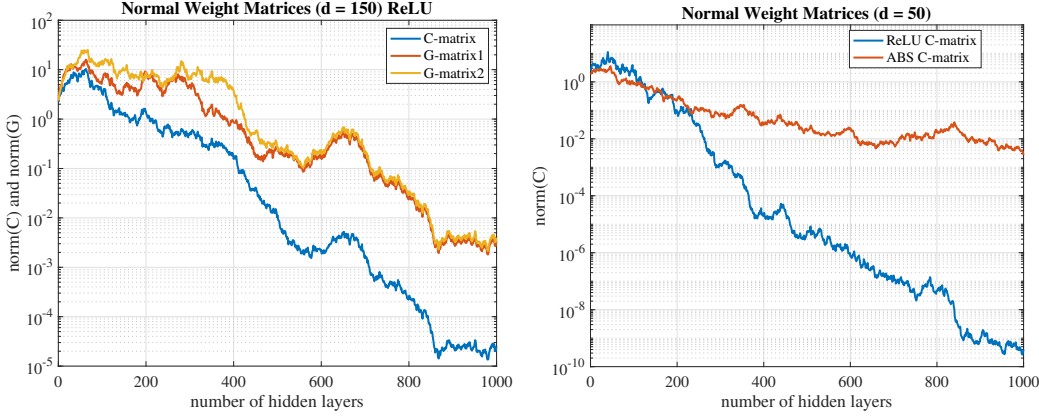

Figure 6: Sizes of C- and G-matrices for ReLU and absolute-value activations

The right plot of Figure 6 is for the comparison of two different activation functions, ReLU, and absolute-value. Starting from a random point-pair, we plot the C-matrices computed on a set of random weight-bias pairs as specified with initialization multipliers $\sqrt{2/d}$ and $\sqrt{1/d}$ for ReLU and absolute-value, respectively. The plot shows clearly that ReLU, with considerably smaller C-matrices, is much more prone to the negative influence of C2C than the absolute-value is. At $L = 1000$, the C-matrix for ReLU is more than three orders of magnitude smaller than that for absolute-value.

## B    VARIABILITY VS. TRAINABILITY

As we have seen, for fully connected neural nets with a fixed number of parameters, variability is initially small, due to low activation ratio values, and then it rises to a peak with the growth of the depth; after that, it starts to fall due to the progression of C2C phenomenon. In this section, we present numerical evidence verifying that the pattern of variability change is highly correlated to the performance of corresponding neural nets. This suggests that variability may well serve as a predictive indicator of the performance of suitable neural nets.

### B.1    EXPERIMENT SETTING

Our experiments are done on a styled synthetic model *checkerboard* in which the data points are the 6561 mesh points of an $81 \times 81$ grid over the square $[-1, 1]^2$ in $\mathbb{R}^2$. These mesh points are divided into two sets, one corresponding to 0-labels and another to 1-labels, so that together they form an 8 by 8 checkerboard blocks, as is shown in Figure 7, where each of the 64 squares contains 81 grid points and the surrounding edges contain 1377 points. The blocks take either 0 or 1 (blue or red) label in an alternating pattern, and the surrounding edges all take the 0-label. In essence, we aim to approximate the piecewise linear, non-smooth function shown in the right plot of Figure 7.

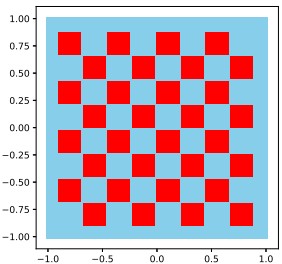 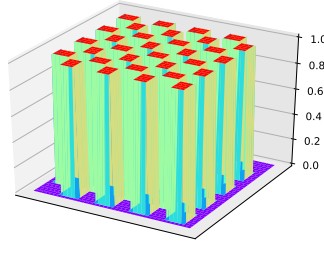

Figure 7: Checkerboard: left plot for data points; right plot for labels; colors match binary labels.

We train fully connected, rectangular neural nets (FCNets) with the same number of nodes or neurons (but see below for exceptions) at each hidden layer. In each of our experiments, when we vary the number of hidden layers $L$, we adjust the width $d$ so that the total number of model parameters (i.e., weights and biases) is kept to a prescribed constant (though, in order to do so we sometimes have to add or delete one or two nodes from some hidden layers). It is worth noting that as the depth grows deeper, the width becomes narrower.

We randomly split the dataset of 6561 points into two parts: $25\%$ as a training set containing $m = 1640$ samples, and the remaining $75\%$ as a testing set. Denoting the training set by $\{x_i\}_{i=1}^m$, we minimize the least squares loss function,

$$\min_{\mathbf{W},\mathbf{b}} f_L(\mathbf{W}, \mathbf{b}) \equiv \sum_{i=1}^m \|\hat{F}_L(x_i; \mathbf{W}, \mathbf{b}) - y_i\|_2^2, \tag{18}$$

where $\hat{F}_L(x, \mathbf{W}, \mathbf{b}) : \mathbb{R}^2 \to \mathbb{R}^2$ is constructed from the neural net function $F_L(x, \mathbf{W}, \mathbf{b})$ in (2) plus an input layer and an output layer, and each label vector $y_i \in \mathbb{R}^2$ is either $(0, 0)^T$ or $(1, 1)^T$, representing

binary labels. We refer to the objective function value as the training loss and the percentage of correctly classified data points as the training accuracy. We note that a 100% training accuracy generally does not imply a 0-training loss.

To ensure that the optimization calculation is done sufficiently, we apply the gradient descent method (instead of SGD) with 40000 iterations without a stopping criterion. For each run, we always try 10 different initial learning rates (step-sizes) as in

$$\{0.001, \ 0.003, \ 0.006, \ 0.01, \ 0.03, \ 0.06, \ 0.1, \ 0.3, \ 0.6, \ 1.0\}$$

and then pick the best result for output. During the 40000 iterations, learning rates are reduced by a factor of 5 three times at the junctures corresponding to iterations 20000, 28000, and 36000, respectively.

We initialize weight matrices and bias vectors with i.i.d. standard normal random numbers, and the weight matrices are then scaled by the Xavier/Kaiming initialization constants. The activation function is ReLU and ABS. All trial instances are run with 10 different samples of random initial parameters. For final output, we always record and output the best result in terms of the training error regardless of where it occurs during training (in the middle or at the end).

## B.2 COMPUTATIONAL RESULTS

We show the results of training FCNets by GD of varying depths (x-axis) and parameters (from left to right in each row). For each network, we compute the mean value of training/testing loss/accuracy. All result is reasonably stable as shown by the std of 10 runs. Once over-parameterization happens, there exists a bottom of the valley in Figure 8 to get or approach to the global optimum.

We fix the total number of parameters at $N_w = 1600, 2400$ and $3200$. For each $N_w$ value, we solve Problem (18) with FCNets-ReLU and FCNets-ABS, following the procedure described in the previous subsection, with the number of hidden layers $L$ varying from 2 to 31 (with increment 1 up to 20 then increment 2 afterwards) for ReLU and extending to 41 for ABS. The results, including training and testing losses and accuracies, are summarized in Figure 8 containing 6 plots in 3 columns corresponding to the 3 values of $N_w$.

As previously shown, when networks grow deeper, variability first increases and then decreases. As we observe from Figure 8, there are remarkably similar patterns in all curves in Figure 8 for either training and test accuracies or losses (for which the rise-and-fall pattern is flipped).

In particular, we compare the left plot of Figure 2 with the third column of Figure 8. For these two cases, the numbers of total model parameters are sufficiently close (3300 vs. 4000) to roughly examine the locations of interests. Indeed, the peak of variability occurs at $L = 12$ in the left plot of Figure 2, while the best performance of the FCNet happens between $L = 10$ and $L = 15$, as can be seen in the third column of Figure 8.

We offer the following interpretations of the experimental results, as pertinent to the relationship between network variability and trainability.

- Variability in the data space indicates a model's expressivity, and in turn, it implies sensitivity in the parameter space. A well-designed and computable measure of variability, which remains a topic of further research, could serve well as an indicator of trainability.

- With low variability, models apparently have more local traps, making training difficult. On the other hand, near or around variability peaks, there appears to exist few or no local traps, as evidenced by the middle sections in the plots of Figure 8 where the training process seems to reach global optima with few or no exceptions, at least so in over-parametrized models.

- As expected, over-parametrization helps trainability. However, it may not always benefit generalizations much or at all, as is evidenced by the curves of testing loss and accuracy in Figure 8.

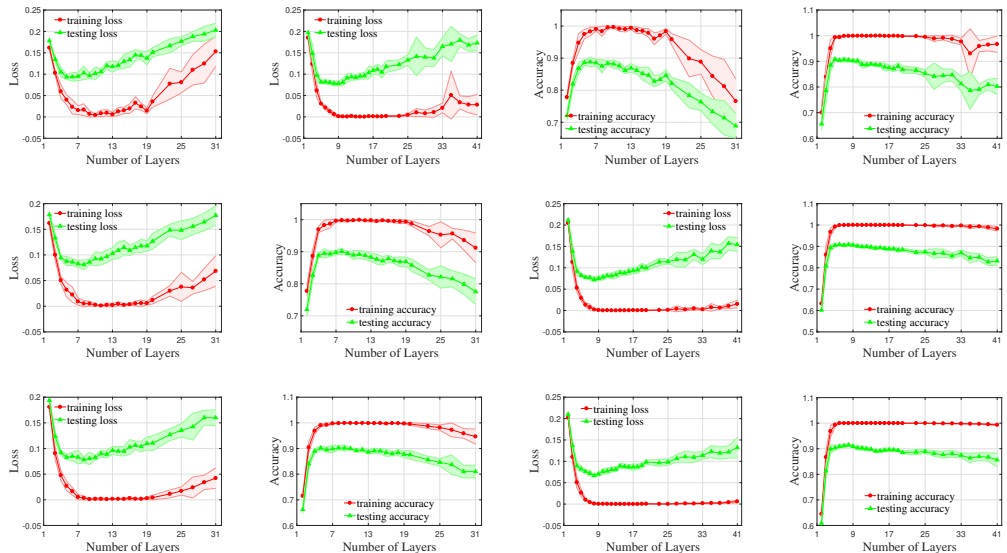

Figure 8: **FCNet-ReLU vs FCNet-ABS on Checkerboard.** The 1st-row contains computed training and testing losses, and the 2nd-row training and testing accuracies. Columns 1 to 3 (from left to right) are for results corresponding to the number of model parameters equal to 1600, 2400, and 3200, respectively (recall that the training set contains 1640 samples). Each group of curves depicts the mean and variance of 10 random runs.

- FCNet-ReLU is unable to reach near-zero loss values with more than 20 hidden layers, but FCNet-ABS still succeeds even after the hidden-layer number exceeds 30, confirming that ABS works better than ReLU in deeper DNNs.

## C   HANNET

### C.1   HANNET ALGORITHM

---

**Algorithm 1** HanLayer.

---
**Input**: vector $x$ and parameters $(u, b)$.
**Output**: $y = \text{HanLayer}(x; u, b)$
1: **function** HANLAYER($x, u, b$)
2:     $z = x - 2\frac{u^\intercal x}{\|u\|^2}u + b$           ▷ Householder Reflection plus bias
3:     $y = \text{ABS}(z)$           ▷ Absolute-value Function Activation
4:     **return** $y$
5: **end function**

---

### C.2   MIXER BLOCK

Figure 9 presents two Mixer structures using FC-Layers and Han-Layers.

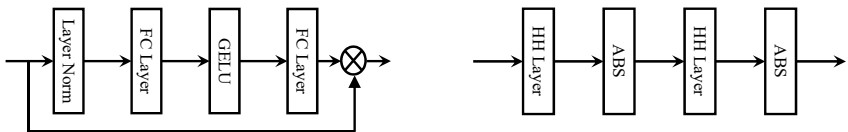

Figure 9: Two implementations of the channel mixing module using MLP-Mixer and Han-Mixer, respectively. HH denotes Householder.

## D  EXPERIMENT

### D.1  EXPERIMENT SETTINGS

Our Checkerboard dataset is similar to the one in Section 4 but more complicated, shown as Figure 4 with 6561 mesh points in a $12 \times 12$ block checkerboard over the square $[-1, 1]^2$. We consider, and our experiments have confirmed, that this dataset is a highly challenging one for standard DNNs.

|  | Datasets | dim | Training set $N$ | Testing set $N$ |
|---|---|---|---|---|
| Synthetic | Checkerboard | 2 | 1640 | 4921 |
| Regression | Elevators | 18 | 3320 or 13279 | 13279 or 3320 |
|  | Cal Housing | 8 | 4128 or 16512 | 16512 or 4128 |
| Image Classification | CIFAR-10 | 3072 | 50000 | 10000 |
|  | CIFAR-100 | 3072 | 50000 | 10000 |
|  | STL-10 | 27648 | 5000 | 8000 |
|  | ImageNet32 | 3072 | 1281167 | 50000 |

Table 4: Dataset statistics: $N$ is the number of samples and dim the dimension of data vectors. We use the mean-square error loss function for the checkerboard and the two regression datasets, and the cross-entropy loss function to classify images. We use stochastic gradient descent (SGD) with momentum to train Checkerboard. To solve each test instance, we run SGD using each of the following 10 initial learning rates:

$$\{0.001, 0.005, 0.01, 0.025, 0.05, 0.075, 0.1, 0.25, 0.5, 1\}$$

and select the best result as the output. In addition, the initial learning rate is annealed by a factor of 5 at the fractions 1/2, 7/10 and 9/10 of the training durations. Other parameter settings for SGD are listed in Table 5 below.

| Dataset | Iteration | batch size | weight decay | momentum |
|---|---|---|---|---|
| Checkerboard | 40000 | 100 | 0.0 | 0.9 |

Table 5: SGD parameters.

For the two regression datasets, we choose Adam (Kingma & Ba, 2015) as the training method which seems to be the method of choice for several works in that area including (Tsang et al., 2018). For these image classification datasets, we choose a modified Adam called AdamW (Loshchilov & Hutter, 2018). Detailed settings are in Table 6 below.

All the training is done using PyTorch (Paszke et al., 2019) running on a shared cluster. Finally, we mention that we always run multiple times for each test instance, starting from different random initializations of model parameters. All the reported values are the average of at least 5 runs. It is noted that the results in each block on Figure 5 are the average of 9 runs. We randomly manufacture the train set 3 times and do 3 trials on each set.

| Dataset | Optimizer | epochs | lr | batch size | weight decay |
|---------|-----------|--------|-------|------------|--------------|
| Elevators/ Cal Housing | Adam | 300 | 0.001 | 100 | 0.0 |
| CIFAR-10/ CIFAR-100 | AdamW | 600 | 0.001 | 256 | 0.1 |
| STL-10 | AdamW | 300 | 0.001 | 64 | 0.1 |
| Imagenet-32 | AdamW | 300 | 0.001 | 512 | 0.1 |

Table 6: Adam/AdamW parameters.

## D.2 ABLATION STUDY ON CHECKERBOARD

| | Layer type | | Activation type | | Test accuracy |
|-----|:---:|:---:|:---:|:---:|:---:|
| | H | FC | ABS | ReLU | |
| (a) | ✓ | ✗ | ✓ | ✗ | 99.2% |
| (b) | ✓ | ✗ | ✗ | ✓ | 66.2% |
| (c) | ✗ | ✓ | ✓ | ✗ | 85.3% |

Table 7: Ablation study on a $100 \times 17$ network framework: Effects of layer and activation types on performance. H denotes Householder and FC denotes fully-connected layers.

To better grasp how much the two HanNets components, Householder weighting and ABS activating, contribute to the surprising results on the Checkerboard dataset, we conduct an ablation study with three different configurations in a $100 \times 17$ network framework. The results are listed in Table 7 where the layer type is either Householder (H) or fully connected (FC) and the activation type is either ABS or ReLU. Table 7 suggests that Householder weighting and ABS activating be equally critical to the 99% generalization accuracy of the HanNet. We investigate the heavily lacking of variability in the networks with Householder matrix plus ReLU function affects the model performance.

## D.3 FLIPPING TRAINING LABELS ON CHECKERBOARD

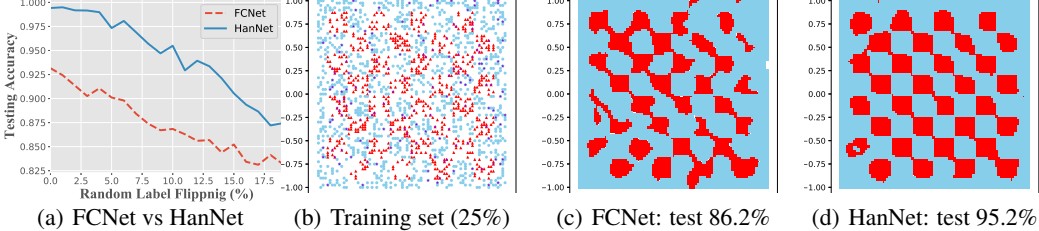

(a) FCNet vs HanNet    (b) Training set (25%)    (c) FCNet: test 86.2%    (d) HanNet: test 95.2%

Figure 10: (a) Testing accuracy in FCNet and HanNet. (b) Training set: violet-colored points are those whose 10% labels are flipped (background color represents their original label). (c) FCNet: the best testing accuracies are 86.5% in the model trained from (b). (d) HanNet: the best testing accuracy is 95.2% .

To check the stability of the near-perfect result of the HanNet, we randomly flip training labels from 0 to 20% on Problem 4. Figure 10 (a) presents the testing accuracy in HanNet and FCNet, where the network architecture is $100 \times 17$ for HanNet and $100 \times 6$ for FCNet. The training result for FCNet is given in Figure 10 (c), while Figure 10 (d) are, the best results for the HanNet in terms of testing accuracy. The best result shows that HanNet can still learn well the original pattern from seriously damaged labels.

## D.4 REGRESSION DATASETS

Table 1 presents the model specifications and statistics of datasets (Elevators and Cal Housing). Figure 11 shows the NRMSE values on two FCNets and HanNet.

| Model | Depth × Width | Elevators | | Cal Housing | |
|---|---|---|---|---|---|
| | | Parameters | AR(%) | Parameters | AR (%) |
| FCNet1 | 5 × 50 | 11201 | 2.23 | 10701 | 2.33 |
| FCNet2 | 5 × 200 | 164801 | 0.60 | 162801 | 0.61 |
| HanNet | 20 × 200 | 11601 | 34.47 | 9601 | 41.66 |

Table 8: Model specifications and statistics of datasets (Elevators and Cal Housing)

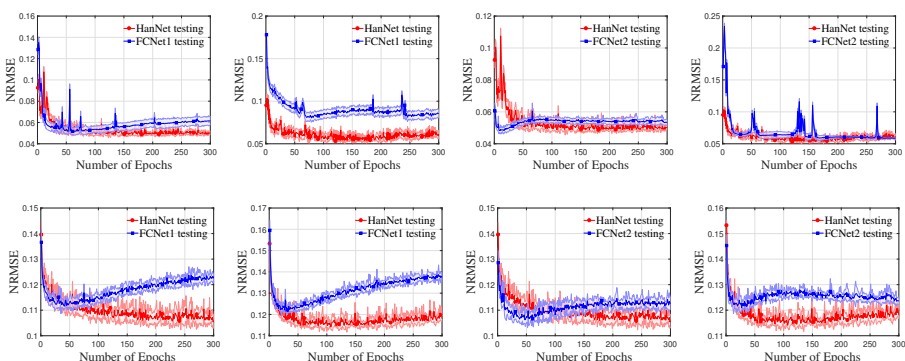

Figure 11: All figures show testing performance where the top row is for the Elevators dataset; and the bottom row for Cal Housing. From left to right: (1) HanNet (red) vs. FCNet1 (blue) on 80% training data, (2) then on 20% training data; (3) HanNet (red) vs FCNet2 (blue) on 80% training data, (4) then on 20% training data. Note that HanNet testing results are repeated.

## D.5 SETTINGS ON IMAGE CLASSIFICATION DATASETS

We summarize various configurations of each block on different data sets in Table 9. We run Mixers using the following number of layers {1, 2, 4, 8, 12, 16} and select the best one on each dataset. We use the convolution stem recommended by (Xiao et al., 2021) instead of the one in (Tolstikhin et al., 2021). Our convolutional stem designs use five layers, including four 3 × 3 convolutions and a single 1 × 1 final layer. The output channels are [64, 128, 256, 512, 512] on CIFAR10, CIFAR100 and STL10, and [128, 256, 512, 1024, 1024] on ImageNet32.

| | CIFAR | STL10 | ImageNet32 |
|---|---|---|---|
| Patch size | 4 × 4 | 8 × 8 | 4 × 4 |
| Sequence Length | 64 | 144 | 64 |
| Channel number | 512 | 512 | 1024 |

Table 9: The architecture on one MLP-block or Han-block. For simplification, in our experiments, all hidden MLP weights are square matries.

## E PLOTS FROM LANDSCAPE EXPERIMENTS

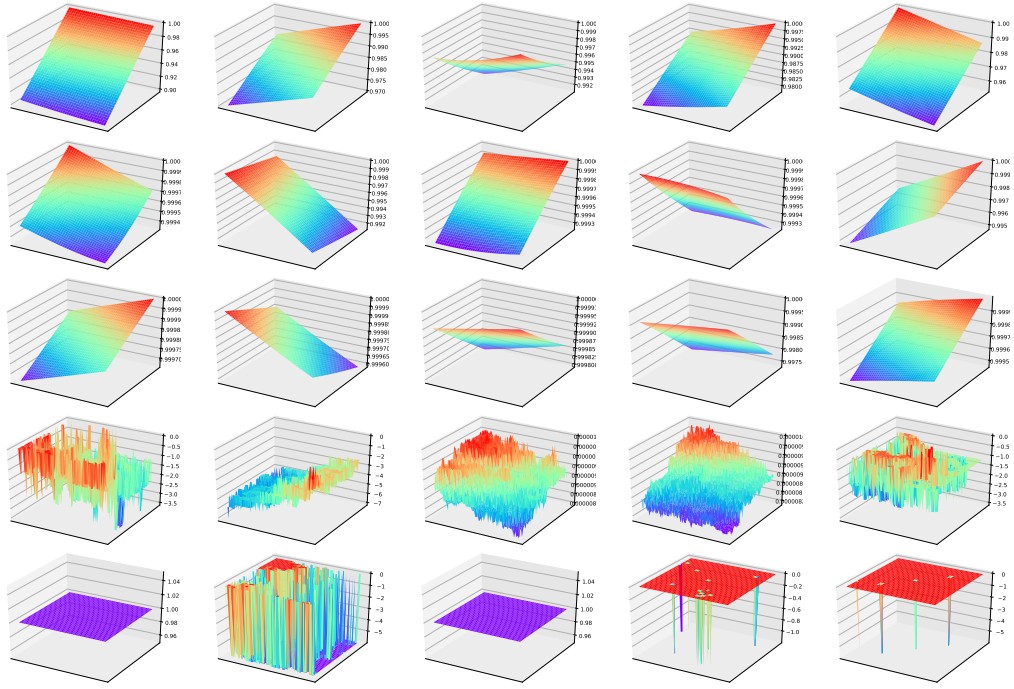

Figure 12: Visualization of variability on FCNet-Sigmoid with 2,4,6,10,15 hidden layers.

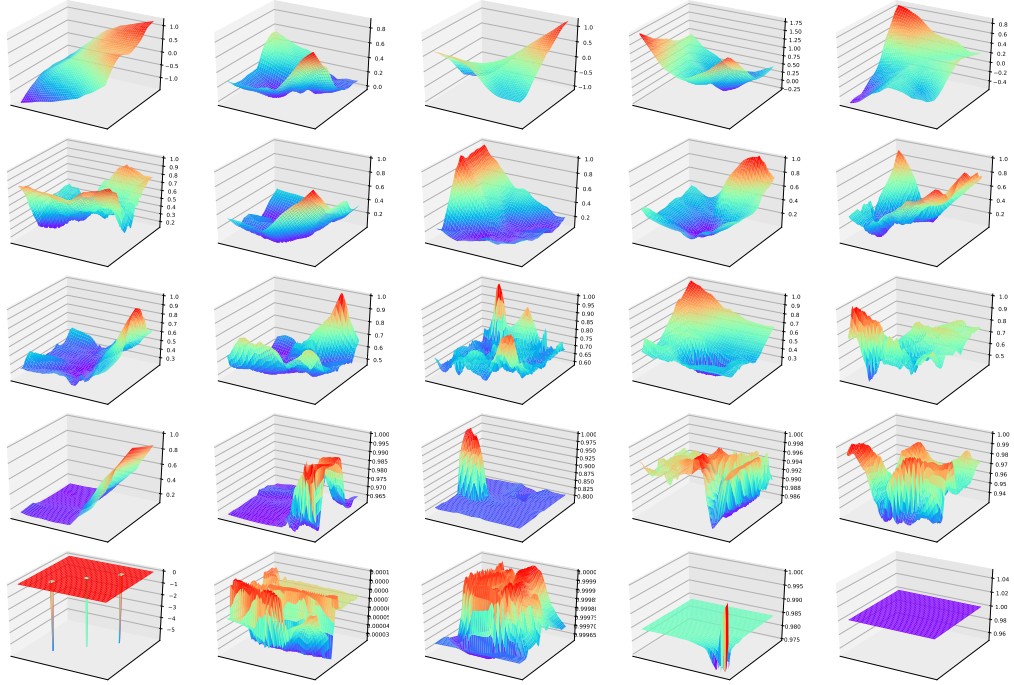

Figure 13: Visualization of variability on FCNet-ReLU with 2,10,20,40,60 hidden layers.

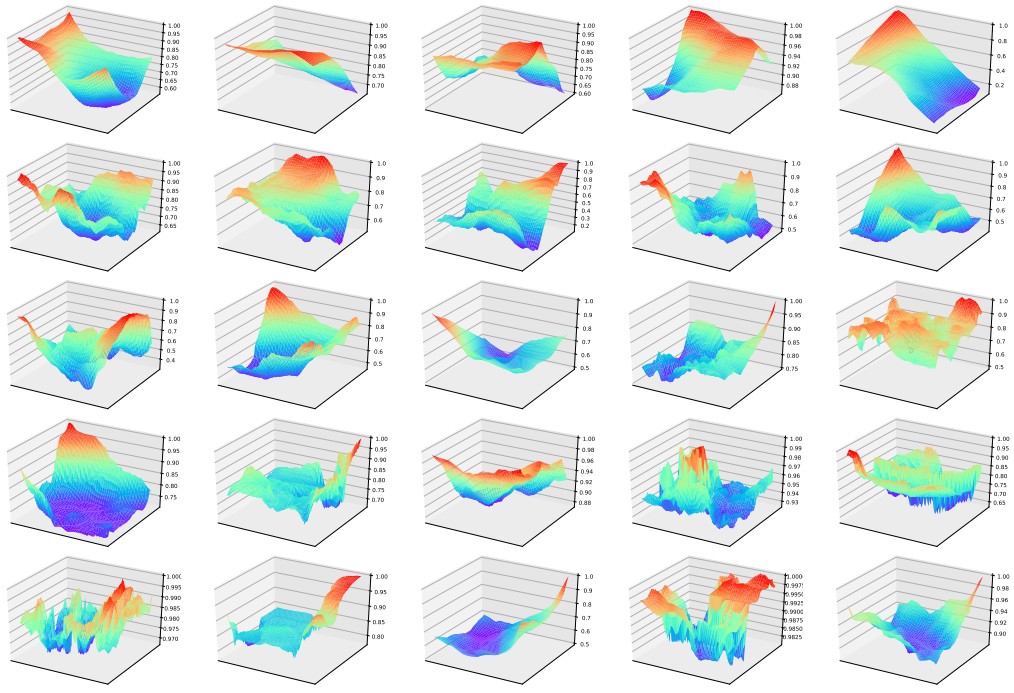

Figure 14: Visualization of variability on FCNet-ABS with 2,10,20,40,60 hidden layers.

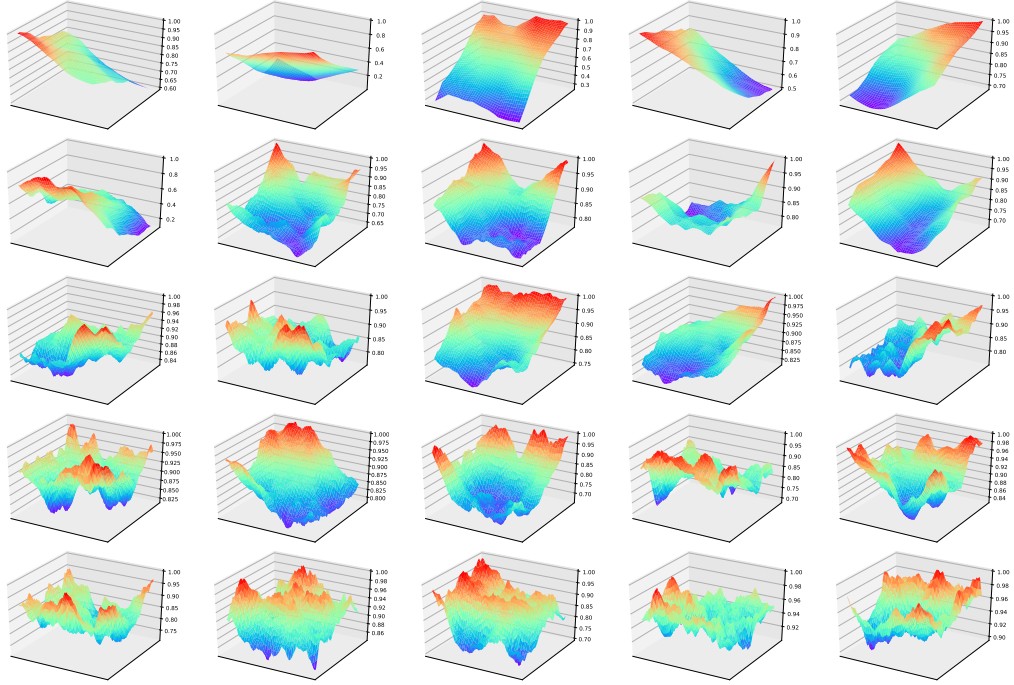

Figure 15: Visualization of variability on HanNet with 2,10,20,40,60 hidden layers.

