# OpenReview forum: "Variability of Neural Networks and Han-Layer: A Variability-Inspired Model"
_ICLR.cc/2022/Conference — ICLR 2022 Submitted_

### Official Review · Reviewer_7Vdi · 2021-11-01

**Correctness:** 2
**Technical Novelty And Significance:** 2
**Empirical Novelty And Significance:** 2
**Recommendation:** 3
**Confidence:** 4

**Main Review:**

__Clarity:__ The paper is clearly written and easy to follow.

__Comments:__
- The __variability__ is only loosely defined as "richness of landscape variations". It seems that it is understandable but a more accurate definition is needed.
- The connection of the above definition to equation 4 (measurable property) is not clear. The only explanation provided in the paper is "seems to have worked well with two-dimensional data spaces". Which is not specific.
- Equation 4 does not specify the distribution over W and b. Paper is silent about that except "for randomly sampled (W, b) with proper scalings". That does not specify the form of distribution.
- The motivation for the proposed _Han-layer_ seems to be based only on the nice behavior of equation 4 resented in Figure 2. That does not look solid for me.
- The empirical improvements are tested only on MLP-Mixer models. While all the datasets are different from those used in the original MLP-Mixer paper.

__Conclusions__:
- _Variability_ is indeed an interesting concept, but it is neither explained clearly enough nor compared accurately with other generalization measures.
- _Han-layer_ does not have a solid connection to _variability_.
- The empirical results are nice but do not seem to be solid enough in isolation from other contributions (e.g., tested only on one architecture, gains are relatively small).

The motivation (__The purpose of this work is to gain more insights into the behaviors of DNNs and then use them to build new DNN models.__) provided in the work is nice. However,  my feelings are that the paper delivers on the opposite. It proposes a new layer and uses it to build new variants of MLP-Mixer. While all the parts of the work feel half-baked.

**Summary Of The Paper:**

The paper introduces a new notion of _variability_ of a DNN, that intuitively is the "richness of landscape variation" wrt data and parameters. The paper attributes the _variability_ to the generalization performance of fully connected DNNs. Inspired by the increasing _variability_ the work proposes _Han-layer_ that is a linear layer with a Householder matrix and abs(.) activation function. The empirical results show that _Han-layer_ can improve the performance of MLP-Mixer on various datasets.

**Summary Of The Review:**

The paper aims to propose new a measure associated with generalization, use it to define new models, and gain some insights into the behavior of DNNs.  However, all the parts of the work feel half-baked.

---

> ### Author Response · Authors · 2021-11-22
> **Thank you for your careful review of our manuscript.**
>
> Thank you for your careful review of our manuscript.
>
> Q.1 The variability is only loosely defined as "richness of landscape variations". It seems that it is understandable but a more accurate definition is needed.
>
> Variability is essentially a qualitative concept that is subject to interpretations according to context. A precise definition is not needed nor plausible.
>
> Q.2 The connection of the above definition to equation 4 (measurable property) is not clear.
>
> The reason for choosing the third derivative is as follows. The landscape function is a quadratic function of the network output. A linear output would result in zero third derivatives. We consider linear outputs to be a lack of landscape variations, therefore using third derivatives.
>
> Q.3 Equation 4 does not specify the distribution over W and b.
>
> We initialize and scale parameters (W,b) by standard methods including Xavier or Kaiming Initialization.
>
> Q.4 The motivation for the proposed Han-layer seems to be based only on the nice behavior of equation 4 resented in Figure 2. That does not look solid for me.
>
> We respectfully disagree with your view.
>
> Q.5 The empirical improvements are tested only on MLP-Mixer models.
>
> Han-Layers are proposed to be used in place of FC-Layers, so we use the MLP-dominant model, MLP-Mixer, to evaluate performance.

---

> > ### Comment · Reviewer_7Vdi · 2021-11-23
> > **Response**
> >
> > Dear authors,
> >
> > Thank you for your clarifications, and I extremely apologize for the late reply.
> >
> > TL;DR: My questions are not sufficiently addressed. So I hold the score.
> >
> > To authors:
> > - I encourage authors to fit the work into a large body of generalization measures research (compare with others, etc.).
> > - I potentially can see the variability as useful property to understanding generalization.
> > - Also hope for more ideas about the generalization of measuring variability to big models.
> >
> > > Variability is essentially a qualitative concept that is subject to interpretations according to context. A precise definition is not needed nor plausible.
> >
> > At this point, I still see it as a concern. It is hard to understand and measure something that is not defined. Also, I would like to highlight that "more accurate" != "precise".
> >
> > > The reason for choosing the third derivative is as follows. The landscape function is a quadratic function of the network output. A linear output would result in zero third derivatives. We consider linear outputs to be a lack of landscape variations, therefore using third derivatives.
> >
> > Makes sense, thank you!
> >
> > > We respectfully disagree with your view.
> >
> > Very interested in hearing an extended response.
> >
> > > Han-Layers are proposed to be used in place of FC-Layers, so we use the MLP-dominant model, MLP-Mixer, to evaluate performance.
> >
> > The part of the question is not reflected in the answer "While all the datasets are different from those used in the original MLP-Mixer paper."

---

> > > ### Author Response · Authors · 2021-11-24
> > > **Response to Reviewer 7Vdi**
> > >
> > > Thanks for your useful suggestions of our manuscript.
> > >
> > > We thank the reviewer for pointing out that we can compare variability to other generalization measures. Our work on these conceptual notions is certainly still preliminary. Calculating variability accurately in high dimension space is still a challenging problem.  Due to insufficient computing resources, we do not run the dataset in the original MLP-Mixer paper. Current experimental results show that our model can improve model performance in many cases. To us, the potential benefits of Han-layer have been made evident.

---

### Official Review · Reviewer_QXHA · 2021-11-02

**Correctness:** 3
**Technical Novelty And Significance:** 3
**Empirical Novelty And Significance:** 2
**Recommendation:** 5
**Confidence:** 3

**Main Review:**

[Strengths]
- This paper is very well written and easy to follow.
- The synthetic experiments are well designed and visualized, and the corresponding results and conclusions are interesting.

[Weaknesses]
1. On the new concepts
- The newly proposed concepts are inspiring but not rigorous, thus producing more questions than it answers.
- Variability seems to have a close relationship with the expressiveness of a neural network. Since few theoretical or rigorous discussions are provided for variability, it remains conceptual and abstract except for the visualization in Figure 2. Moreover, as stated by the authors, it is not the case that the higher variability the better. Then what could be the practical use of the proposed concept?
- For the activation ratio, discussions are even less sufficient. For example, keeping the number of activations to be the same, then reducing network parameters would improve AR. Suppose we reduce the network parameters by introducing low-rankness to the model weights. Then AR increases as rank reduces, and there seems to be a trade-off between parameter efficiency and network expressiveness. Can AR contribute any more precise / more widely-applicable insight?

2. On the synthetic experiments
- The landscape experiments compared FCNets with HanNet. However, it is well-known that deep FCNet has the gradient vanishing problem, and thus it is not surprising that the collapse-to-constant phenomenon exists. Currently, Figure 1 shows that HanNet is a solution to this problem. However, existing solutions are mostly based on residual connections and batch normalization. Shouldn't the solutions be compared against each other? That is, what does the landscape look like for FCNets with residual connections and/or batch normalization?

3. On the image classification datasets
- The datasets and models are too small to shed light on the effectiveness of the proposed method in the more practical scenarios. Because Han-layer has only a few parameters, it is possible that Han-layer is beneficial for toy examples but not on much more complicated real applications.
- Scaling down the resolution of ImageNet samples makes training less costly but this also makes it difficult to compare the proposed method to existing benchmarks. Thus, it is difficult to judge the significance of the improvements. Besides, WideResNet could be a bit outdated. For a more SOTA ResNet-based result, authors may refer to

Bello, I., Fedus, W., Du, X., Cubuk, E. D., Srinivas, A., Lin, T. Y., ... & Zoph, B. (2021). Revisiting ResNets: Improved training and scaling strategies. NeurIPS 2021.

[Other details]
- typo in Section 3.1: "rounded to the nearest integetr"
- Since geometric mean is actually used for equation (4) instead of the arithmetic mean, which is the sample version of expectation, I recommend changing the expectation to a notion of the geometric mean.

**Summary Of The Paper:**

This paper introduces some concepts, i.e., variability, activation ratio, and collapse to constants, to help understand deep neural networks, and proposes a new layer called Householder-absolute layers to improve network variability.
Experiments are conducted on both synthetic datasets and empirical datasets. The synthetic experiments show some interesting results and visualizations, and for the real datasets, improvements over baseline architectures can be observed.

**Summary Of The Review:**

This paper is easy to read and provides some interesting discussions and visualizations. However, concerns exist regarding
- the usefulness of the proposed concepts;
- the design of the landscape experiments; and
- the effectiveness of the proposed model on more complicated applications.

---

> ### Author Response · Authors · 2021-11-22
> **Thank you for your careful review of our manuscript.**
>
> Thank you for your careful review of our manuscript. Here, we provide feedbacks to your review.
>
> Q.1 On the new concepts.
>
> In this paper, we introduce a new angle to view DNN that we called variability, as well as a couple of related notions such as Activation Ratio (AR) and Collapse to Constant (C2C). Variability reflects levels of landscape variations of network functions, while C2C is a consequence of the loss of variability. Admittedly, variability is difficult to quantify in general, and we propose a computable measure for low-dimensional cases. We characterize C2C by C-matrix which is distinct from the usual back-propagation matrix (denoted as the G-matrix) that characterizes gradient problems. Our work on these conceptual notions is certainly still preliminary, but it does offer new and different perspectives from familiar notions such as gradient vanishing or dead neurons. In fact, C2C phenomenon can happen not only to ReLU but also to all ReLU-like activation functions.
>
> The variability viewpoint has inspired us to construct a new architecture called Han-Layer to increase AR and reduce the chance of C2C.
>
> Q.2 On the synthetic experiments.
>
> We observe FCNet-ReLU + Batch normalization + residual connection is more resistive to C2C than FCNet-ReLU in our landscape experiments. However, on the Checkerboard dataset, our experimental result shows that the testing accuracy is still about 85% in FCNet+BN+RS.
>
> Q.3 On the image classification experiments.
>
> Though ImageNet32 is a down-sampled version, it could be a more difficult problem than ImageNet itself. Our model can improve model performance in many cases. Maybe on real applications, parameters from a Han-layer only model are not enough to fit all data points, but we can combine Han-Layers with other layers (FC-Layers or convolution layers). For example, the result in ImageNet32 shows that Han/MLP-Mixer combination model outperforms pure MLP-Mixers.

---

> > ### Comment · Reviewer_QXHA · 2021-11-30
> > **Response**
> >
> > Thank you very much for the reply.
> >
> > I recognize that the ideas in this paper are interesting and could be inspiring, but there are necessary improvements which should not be left as future work / to other people. Thus, I am keeping my original ratings.

---

### Official Review · Reviewer_8jvQ · 2021-11-02

**Correctness:** 2
**Technical Novelty And Significance:** 2
**Empirical Novelty And Significance:** 2
**Recommendation:** 3
**Confidence:** 4

**Main Review:**


**Strengths**
- Notion of variability is novel.
- Han-layers outperform fully connected layers on variety of tasks with lesser parameters.
- Han networks have better variability than typical fully connected layers.


**Weaknesses**:


- Use of Householder matrix is not novel and they have been used in the RNNs as well as DNNs. For ex.  https://arxiv.org/pdf/1803.09327.pdf  uses Householder matrices to construct MLP equivalents (see Spectral-MLP and Spectral-Resnet architectures).

- Although Householder matrices mitigate the vanishing or exploding gradient issues, the added non-linearity and higher depth, would mean additional training time and inference cost. As such a trade off exists, and that has not been explored in this work.

- Parameter count is one way to compare architectures, but more importantly one should add additional metrics such as inference time, floating point operations, training time, etc. This is simply due to the fact that parameter count alone is somewhat misleading. Consider the case of convolutional operations vs matrix multiplication. The number of parameters in the convolution would be very small compared to a similar sized FC operation, while the FLOPs / compute required for the convolution would in general be much higher than the FC operation. At the very least, one should FLOPs comparison for MLPMixer and MLPMixer+HanMixer operations for architecture resulting in similar performance.


- One important critic of the variability definition is that it uses third order gradient information and this would easily fail to scale to even problems where the DL community has tried to use second order gradient information. It would be good to explain the reasoning for this choice of the variability and did the authors consider any alternatives to this?


- Increasing Activation Ratio is not something one should promote in general. Point being, when two models with same parameters are compared, if one has more activations than the other, this architecture will be cost more compute. Increasing AR can only be promoted when you have cheap activation functions, for ex. it would be much more costly to evaluate an exponential unit than a ReLU.


- One trend that seems to be constant in the empirical evaluations is the substantially increased depth of the network. Are there scenarios where these deep networks are cost efficient? Since any increase in depth comes with added overhead of working memory and inference time due to the sequential dependency on the hardware.

- Its somewhat clear from the paper that Han-layers do loose some of the representational power which the FC layers have, in the sense that to achieve similar results the Han-networks require more depth.



**Questions for Authors**

- In Appendix D.4, Fig.11, baseline graphs (blue lines) have a trend where they reach a minima very quickly and then suddenly keep on increasing forever. Is there any explanation for this behaviour? Were the baselines implemented corectly? Are there any difference in the experimental setups such as learning rate, weight decay, optimizer for baseline and HanNet?

- Have you compared your architecture with the baselines in terms of metrics such as training time, inference time, floating point operations, working memory usage, etc.? Is there any reason why these metrics have been omitted in the paper?

- In Sec.3.1, Why draw random networks in this space? In reality, one should expect some structure in the data and similarly the networks should be learnt to explore such structure.

- Why would one not expect the sigmoid plot in the Figure~1 to be flat? Is it not possible that simply due to the fact that the network weights are initialized randomly, the activation is mostly dead or operating in a saturated regime?




**Writing Clarity**:
- Spelling error: Sec.3.1, para 2, ... nearest integetr..
- Simplification: Sec.4.3, para 1, so-called G_L matrix is nothing but the gradients of the function w.r.t. parameters.
- Introduction is very hard to understand due to lack of following definitions. Since these are the basic issues the paper is dealing with, one would expect simple and intuitive definitions in the introduction. (a) Does not explicitly define variability, (b) Does not define collapse to constant.



**Summary Of The Paper:**

**Contributions**:
- Introduces the notion of variability to explain the better trainability and generalization of certain deep neural architectures. This is backed by a concrete definition of the variability in terms of the third order derivative information of the loss landscape.

- Empirical experiments verify that variability correlates positively to the number of activations and negatively to a phenomenon called Collapse to Constants (C2C).

- Inspired by the notion of variability, authors propose Householder-absolute neural layer aka Han-layer that replaces square weight matrices with Householder reflectors, and use absolute-value function for activations.

-  HanNet achieves a high variability as well as guarantee an immunity to vanishing or exploding gradients, and chance for collapse to constants has been diminished.

- HanNets achieve better generalization with lesser parameters than fully connected networks.



**Summary Of The Review:**

Notion of variability maybe a step in the right direction to explain generalization and trainability of DNNs. While the Han-layers do seem to outperform FCNets, many questions remain unanswered in this work, namely,
- Are Han-nets compute hungry (train, inference) when compared to FCnets?
- Is the additional depth necessary?
- Is it harder to train Han-nets in comparison to FCnets, due to orthogonality and additional non-linearity?
- Are there simpler definitions of the notion of variability that extend to higher dimensions than the toy examples illustrated in the paper?

---

> ### Author Response · Authors · 2021-11-22
> **Thank you for your careful review of our manuscript.**
>
> Thank you for your careful review of our manuscript.
>
>
> Q.1 Use of Householder matrix is not novel and they have been used in the RNNs as well as DNNs.
>
> We propose a novel model, HanLayer, composed of a householder matrix followed by ABS activation function, while the previous works use several Householder matrices to replace a dense weight matrix W.
>
> Q.2 The use of the 3rd derivative.
>
> The reason for choosing the third derivative is as follows. The landscape function is a quadratic function of the network output. A linear output would result in zero third derivatives. We consider linear outputs to be a lack of landscape variations, therefore using third derivatives.
>
> Q.3 Have you compared your architecture with the baselines in terms of metrics such as training time, inference time, floating-point operations,
>
> Han-Layer reduces the floating-point operation complexity by one order of magnitude from that of FC-Layer. Additionally, normalization methods and/or residual connections are unnecessary for our Han-Layer, which also reduces computation complexity, though optimizing Han-layer computations at the GPU level in PyTorch seems necessary to take advantage of the low complexity.
>
> Q.4 In Appendix D.4, Fig.11, baseline graphs (blue lines) have a trend where they reach a minima very quickly and then suddenly keep on increasing forever. Is there any explanation for this behaviour?
>
> It may be due to overfitting.
>
> Q.5 In Sec.3.1, Why draw random networks in this space? In reality, one should expect some structure in the data, and similarly, the networks should be learnt to explore such structure.
>
> Learning starts from random networks (well-scaled in parameter space). If a network has little variability to begin with, then expected results of learning would not be promising.
>
> Q.6 Why would one not expect the sigmoid plot in the Figure 1 to be flat?
>
> The reason may be that the sigmoid is operating mostly in saturated regimes.
>
> Q.7 Is it harder to train Han-nets in comparison to FCnets, due to orthogonality and additional non-linearity?
>
> We use the same training settings for all models in our paper, such as epochs, batch size, learning rate scheduler, and optimizer. HanNets does not need extra hyperparameter fine-tuning. However, due to high nonlinearity, in some cases, the convergence behavior of HanNets could be more unpredictable than that of FCNets (when without batch normalization).
>
> Q.8 Are there simpler definitions of the notion of variability that extend to higher dimensions than the toy examples illustrated in the paper?
>
> The research of variability is still in its preliminary stage. Admittedly, it is challenging to develop a computable measure for high-dimensional spaces. However, the usefulness of the concept is more in a qualitative sense.

---

### Official Review · Reviewer_xnGa · 2021-11-04

**Correctness:** 2
**Technical Novelty And Significance:** 1
**Empirical Novelty And Significance:** 2
**Recommendation:** 3
**Confidence:** 4

**Main Review:**

Strengths
========
The parametrization of weight matrices as Householder reflectors makes sense as a way to reduce the number of free parameters, and force orthogonality, like was demonstrated in a different context (recurrent networks) by Mhammedi et al. (2017).

The HanMixer layer indeed seems to help improving performance of the CNN stem on small-resolution image datasets.

Weaknesses
==========
The theoretical justification or inspiration is tenuous and flimsy for many reasons.
- The intuition of section 3.1, Fig. 1, and Appendix E comes from a 2D to 1D mapping, which is not a usual setting for neural networks, and is probably not representative of high-dimensional geometry (which can be deceptive).
- Even the definition of "variability" in Eq. (4) does not scale well with the dimension of the input space, so it would be hard to assess its validity (although no attempts are made).
- That definition depends on a distribution over the parameters W and b, which is arbitrary.
- The use of the 3rd derivative is not explained or justified. It is strange as, for instance, linear models are somewhat expressive, and well trainable, though their variability would be 0.
- Empirical correlations in the 2D case between depth and that measure (for a given number of free parameters) does not mean the "activation ratio" is a meaningful measure.
- "variability" is qualified as a surrogate of "expressivity" as defined by Gühring et al. (2020), but expressivity refers to which functions can be expressed by a given class of function approximators, or how "wide" that class is. However, "variability" is then compared to the ease of training / optimizing a network, which would correspond to the training or optimization error of Gühring et al.
- the so-called "collapse-to-constant" phenomenon is not anything new, especially for activation functions like sigmoid (I believe it was addressed by Glorot & Bengio (2010)), and the issue of "dead neurons" with ReLU has been studied. Activation functions less prone to that issue should have been compared: tanh, ELU... Comparing with established techniques like batch normalization and layer normalization could have been interesting, instead of jumping to the conclusion that absolute value was the answer.
- In general, there is no evidence that the specific heuristics and metrics described here generalize beyond the un-trained networks operating on a 2D [-1, 1] grid, or to settings where the total number of parameters (weights) is not fixed.

Experiments are lacking ablations or comparative studies of which elements of the Householder matrix formulation help with the generalization. For instance, orthogonal matrices with more parameters could have been used, or other low-rank approximations with the same number of parameters.

Other
=====
Section 4.1, the statement that "orthogonality is a desirable property for weight matrices in DNNs" requires more discussion, justifications, and citations than "As is well known". I can't find a specific reference, but for instance there has been an argument that reversible/unitary transformations hurt generalization because of the inability to "throw away" irrelevant information.

**Summary Of The Paper:**

The paper introduces an estimator of expressivity or trainability of deep networks, that they call "variability", that characterizes the variations in a scalar output of the network, when the input change, and seems correlated to trainability of the network, and to its depth (to an extent).

The paper proposes to parametrize weight matrices by using Householder reflectors, parametrized by a single vector each, which has two benefits:
- A reduced number of free parameters, which enables more depth (more layers and non-linearities) for the same model size
- Orthogonality of the matrices.

Experiments on a synthetic 2D benchmark, two small-scale regression datasets, and small-resolution image datasets show improved results compared to dense nets.


**Summary Of The Review:**

The parametrization of weight matrices by Householder matrices is interesting, although the experiments are quite preliminary.
However, the whole derivation of proxies for expressivity or trainability is not adequately grounded theoretically, nor demonstrated empirically beyond a small contrived example.
I do not think this paper should be accepted.

---

> ### Author Response · Authors · 2021-11-22
> **Thank you for your careful review of our manuscript.**
>
> Thank you for your careful review of our manuscript.
>
> We will answer the raised questions below.
>
> Q.1 The intuition of section 3.1, Fig. 1, and Appendix E comes from a 2D to 1D mapping, which is not a usual setting for neural networks, and is probably not representative of high-dimensional geometry (which can be deceptive).
>
> We respectfully disagree with your view. Variability is difficult to quantify in general, and we propose a computable measure for low-dimensional cases.
>
> Q.2 Even the definition of "variability" in Eq. (4) does not scale well with the dimension of the input space. / The distribution of W, b is arbitrary.
>
> We have to emphasize that we initialize and scale parameters (W,b) by standard methods including Xavier or Kaiming Initialization.
>
> Q.3 The use of the 3rd derivative is not explained.
>
> The reason for choosing the third derivative is as follows.
> The landscape function is a quadratic function of the network output. A linear output would result in zero third derivatives. We consider linear outputs to be a lack of landscape variations, therefore using third derivatives.
>
> Q.4  Empirical correlations in the 2D case between depth and that measure (for a given number of free parameters) does not mean the "activation ratio" is a meaningful measure.
>
> Thank you for the comment, but we cannot fully agree with the comment. Variability comes from nolinear activation functions.  We construct a new architecture called Han-Layer to increase AR and reduce the chance of C2C.
>
> Q.5 The so-called "collapse-to-constant" phenomenon is not anything new, the issue of "dead neurons" with ReLU has been studied.
>
> For C2C, the dead neuron is a condition where the activation weight is rarely used due to zero gradients. C2C phenomenon is different from the dead neuron since the absolute-value function will never suffer from zero gradient problem but is still not immune to C2C as depth increases. Our paper treats the C2C phenomenon in a most general framework, and our analysis turns out to be a straightforward one. The reason why we choose ABS is that (1). ABS is more C2C-resistive than ReLU is and has more variations in landscape visualization. (2) Along with Householder, ABS helps to ensure that gradients never vanish or explode. (3) ABS is as inexpensive as ReLU.
>
> Q.6 Experiments are lacking ablations or comparative studies of which elements of the Householder matrix formulation help with the generalization.
>
> The reviewer might have overlooked that we have an ablation study on the checkerboard dataset in Appendix D.2.

---

> > ### Comment · Reviewer_xnGa · 2021-11-23
> > **Response**
> >
> > Thanks for your reply. Unfortunately, I'm not convinced by many of your answers.
> >
> > > Variability is difficult to quantify in general, and we propose a computable measure for low-dimensional cases.
> >
> > Proposing a definition and a measure is not the hardest part, the issue is to show it is the right one to use. The article does not make the case that these particular formulations of "variability" and "activation ratio", generalize to other useful settings (or even characterize when they are applicable in practice), as opposed to other measures like condition number, depth, etc.
> >
> > > We have to emphasize that we initialize and scale parameters (W,b) by standard methods including Xavier or Kaiming Initialization.
> >
> > The fact these methods are standard does not mean it is not arbitrary. The usefulness of that metric is even more limited if it is restricted to one or two standard methods.
> >
> > > The reviewer might have overlooked that we have an ablation study on the checkerboard dataset in Appendix D.2.
> >
> > That ablation study only compares the non-linearity used, and Householder vs. full-matrix.
> > It does not address what I mention, which is "*which elements of the Householder matrix formulation* help with the generalization". I explicitly suggested the following, which are not in Appendix D2:
> > * orthogonal matrices with more parameters
> > * other low-rank approximations with the same number of parameters

---

> > > ### Author Response · Authors · 2021-11-23
> > > **Response to Reviewer xnGa**
> > >
> > > We thank the useful comments and suggestions from the reviewer.
> > >
> > > We agree that in our paper, we do not make the case that  "variability" and "activation ratio", generalize to other useful settings. Our work on these conceptual notions is certainly still preliminary. Calculating variability accurately in high dimension space is still a challenging problem.
> > > To us, the new notions in this paper offer new and different perspectives from previous notions, and the potential benefits of Han-layer have been made evident, even though many things remain to be further studied.

---

### Author Response · Authors · 2021-11-22
**Summary of responses**


We thank all reviewers for their valuable comments. First of all, we would like to summarize our work as an overall response to various related criticisms and questions.

In this paper, we introduce a new angle to view DNN that we called variability, as well as a couple of related notions such as Activation Ratio (AR) and Collapse to Constant (C2C). Variability reflects levels of landscape variations of network functions, while C2C is a consequence of the loss of variability. Admittedly, variability is difficult to quantify in general, and we propose a computable measure for low-dimensional cases. We characterize C2C by C-matrix which is distinct from the usual back-propagation matrix (denoted as the G-matrix) that characterizes gradient problems. Our work on these conceptual notions is certainly still preliminary, but it does offer new and different perspectives from familiar notions such as gradient vanishing or dead neurons. In fact, C2C phenomenon can happen not only to ReLU but also to all ReLU-like activation functions.

The variability viewpoint has inspired us to construct a new architecture called Han-Layer to increase AR and reduce the chance of C2C. Han-Layer also reduces the floating-point operation complexity by one order of magnitude from that of FC-Layer. Current experimental results show that our model can improve model performance in many cases when used in place of FC-Layers with far fewer parameters. To us, the potential benefits of Han-layer have been made evident, even though many things remain to be further studied.

Mainly due to space limitation, unfortunately, we did not discuss several related issues in detail that reviewers raised. Among them is the need to optimize Han-layer computations at the GPU level in PyTorch, without which the lower floating-point operation complexity of Han-layers would not fully realize its advantage. In the sequel, we will try to address specific questions.

---

### Decision · Program_Chairs · 2022-01-20

**Decision:**

Reject

**Comment:**

The reviewers generally agreed that the ideas presented in the paper are interesting and novel. However, all reviewers also agreed that the paper is quite preliminary in its current form: the particular approach, while sensible, appears to be somewhat heuristic, and the evaluations are not as complete as necessary to fully evaluate the proposed approach.

Generally, my sense is that there is something quite interesting in this work, but the present paper is too preliminary for publication. I would encourage the authors to take the reviewer comments into account and improve the work into a more complete submission for a future venue.